# Differentiation between *Weissella cibaria* and *Weissella confusa* Using Machine-Learning-Combined MALDI-TOF MS

**DOI:** 10.3390/ijms241311009

**Published:** 2023-07-02

**Authors:** Eiseul Kim, Seung-Min Yang, Dae-Hyun Jung, Hae-Yeong Kim

**Affiliations:** 1Institute of Life Sciences and Resources, Yongin 17104, Republic of Korea; eskim89@khu.ac.kr (E.K.); ysm9284@gmail.com (S.-M.Y.); 2Department of Food Science and Biotechnology, Kyung Hee University, Yongin 17104, Republic of Korea; 3Department of Smart Farm Science, Kyung Hee University, Yongin 17104, Republic of Korea

**Keywords:** *Weissella cibaria*, *Weissella confusa*, MALDI-TOF MS, machine learning, identification

## Abstract

Although *Weissella cibaria* and *W. confusa* are essential food-fermenting bacteria, they are also opportunistic pathogens. Despite these species being commercially crucial, their taxonomy is still based on inaccurate identification methods. In this study, we present a novel approach for identifying two important *Weissella* species, *W*. *cibaria* and *W*. *confusa*, by combining matrix-assisted laser desorption/ionization and time-of-flight mass spectrometer (MALDI-TOF MS) data using machine-learning techniques. After on- and off-plate protein extraction, we observed that the BioTyper database misidentified or could not differentiate *Weissella* species. Although *Weissella* species exhibited very similar protein profiles, these species can be differentiated on the basis of the results of a statistical analysis. To classify *W*. *cibaria*, *W*. *confusa*, and non-target *Weissella* species, machine learning was used for 167 spectra, which led to the listing of potential species-specific mass-to-charge (*m*/*z*) loci. Machine-learning techniques including artificial neural networks, principal component analysis combined with the K-nearest neighbor, support vector machine (SVM), and random forest were used. The model that applied the Radial Basis Function kernel algorithm in SVM achieved classification accuracy of 1.0 for training and test sets. The combination of MALDI-TOF MS and machine learning can efficiently classify closely-related species, enabling accurate microbial identification.

## 1. Introduction

*Weissella* is a genus of lactic acid bacteria that inhabit various sources, such as milk, feces, meat, and vegetables, and currently includes 19 species [1]. Of these, *Weissella cibaria* and *W. confusa* are the dominant species in fresh vegetables and play crucial roles in the fermentation of foods. Studies on *W*. *cibaria* and *W*. *confusa* have mainly referred to exopolysaccharide (EPS) production. EPS production using starter culture is generally preferred in the dairy industry, as it improves bread texture and shelf life in sourdough fermentation [2]. Many researchers have recently determined the health-promoting effects of *W*. *cibaria* and *W*. *confusa* by investigating their probiotic properties [3,4]. *W*. *cibaria* inhibits *Fusobacterium nucleatum*, which causes periodontal disease in the oral cavity [4], and reduces oxidative stress-induced disorders [5]. *W*. *confusa* exhibited high cholesterol elimination, antioxidant activity, and antibacterial activity [3,6]. Despite these favorable characteristics, *W*. *confusa* causes human infections; although the clinical significance of this species is unclear, it mainly causes infections in individuals who are immunocompromised [2]. Thus, this species is usually considered a rare cause of nonfatal human infections and a contaminant. The aforementioned two species are inherently resistant to vancomycin like other *Weissella* species. However, some *W*. *confusa* strains have recently been reported to cause abscesses, bacteremia, and infective endocarditis [7,8].

Several methods have been developed for *Weissella* identification, such as polymerase chain reaction (PCR) assays, sequencing, and matrix-assisted laser desorption/ionization time-of-flight mass spectrometer (MALDI-TOF MS) [9,10,11]. Of these, MALDI-TOF MS has been used in many clinical laboratories for bacterial identification. This method is efficient and convenient for microbial detection, can be performed in minutes with a single colony, and costs only a few US dollars [12]. MALDI-TOF MS can rapidly identify bacteria by generating protein-fingerprint signatures from bacteria cells and comparing them to fingerprints in databases of reference spectra [13]. Therefore, successful bacterial identification using MALDI-TOF MS relies primarily on a database containing the spectra of known organisms [14]. However, commercial databases are unreliable because of a lack of accurate discrimination among closely related species or subspecies, such as *W*. *cibaria* and *W*. *confusa*. Moreover, experimental factors, such as sample preparation, cell lysis method, and instrument performance can affect the quality and reproducibility of MALDI-TOF MS fingerprints [15]. Strains indistinguishable using commercial databases have recently been accurately identified by selecting specific peaks or by machine learning of protein mass spectra obtained through MALDI-TOF MS [16,17].

Recently, machine learning techniques have been increasingly used for microbial identification. Microbial identification is dominated by the application of artificial neural networks (ANN), which involve the use of data resulting from various analytical techniques, such as surface-enhanced Raman spectroscopy, genetic fingerprinting, and infrared spectroscopy [15,18,19]. Most machine-learning studies have focused on the classification of antibiotic-resistant bacteria or the differentiation of pathogenic bacterial species having clinical impact. However, such studies did not include identification schemes for lactic acid bacterial species within a specific taxon. Given the advantages of machine-learning techniques, these techniques could contribute considerably to the microbial taxonomy field [15]. Machine-learning techniques learn from the data and make the most of the information contained in the data. Additionally, recent machine-learning techniques have been utilized to maximize the use of information contained within MALDI-TOF MS, with the ultimate goal of improving species identification [20]. Machine learning can smoothly handle multidimensional data and offer a solution when standard analytical methods fail. Moreover, numerous algorithms have been implemented in various open-source software programs (https://github.com/, 2 July 2023); therefore, any scientist can freely use them.

Here, we evaluated the ability to discriminate among *W*. *cibaria*, *W*. *confusa*, and nontarget *Weissella* species for the first time, by combining MALDI-TOF MS and machine learning techniques. By performing species identification based on similarity searches against the commercial library, the performance of the MALDI-TOF MS database was assessed. The ANN, principal component analysis combined with the K-nearest neighbor (PCA-KNN), and support vector machine (SVM) algorithms were used for classifying *W*. *cibaria*, *W*. *confusa*, and non-target species.

## 2. Results

### 2.1. Molecular Identification

Specific primers for detecting *W*. *cibaria*, *W*. *confusa*, and non-target *Weissella* strains were previously designed through pangenome analysis [9] and used to identify the isolates. Each primer produced an amplification plot for each strain. Of the 125 strains, 44, 46, and 35 strains were identified as *W*. *cibaria*, *W*. *confusa*, and non-target strains, respectively.

### 2.2. Analysis of Mass Spectra with BioTyper Database

In total, 44 *W*. *cibaria* strains, 46 *W*. *confusa* strains, and 35 non-target strains were tested. The BioTyper database (version 2022) included 12 *Weissella* species, including *W*. *cibaria* and *W*. *confusa*, each with four and six spectra, respectively. Consequently, *W*. *cibaria* strains were accurately identified after analysis using BioTyper 3.0 software. High average scores of 2.25 and 2.29 were obtained using the direct transfer-formic and ethanol-formic acid extraction protocols, respectively (Appendix A). Using the aforementioned protocols, *W*. *confusa* strains were also identified with high average scores of 2.00 and 2.02, respectively. However, only five *W*. *confusa* strains were accurately identified (Appendix A). Ten strains having scores between 1.74 and 2.10 were initially identified as *W*. *confusa*. However, upon the second match, they were identified as *W*. *cibaria*, with scores between 1.71 and 2.02. Similarly, 21 strains having scores between 1.74 and 2.13 were initially matched as *W*. *cibaria*. However, upon the second match, they were identified as *W*. *confusa* with scores between 1.72 and 2.12. Ten *W*. *confusa* strains were misidentified as *W*. *cibaria*, with scores between 1.74 and 2.12. When the species could not be accurately identified in the database, additional peak analysis was required for reidentification.

### 2.3. Clustering Analysis of Mass Spectra

The mass spectra of *W*. *cibaria* and *W*. *confusa* showed similar patterns (Figure 1). The mass spectra of each strain for non-target *Weissella* species are shown in Appendix A. We verified whether the mass spectra of MALDI-TOF MS could differentiate among *W. cibaria, W. confusa,* and non-target strains at a statistical level. After all mass spectrometry data were preprocessed, we first analyzed the differences in peak spectra among *W*. *cibaria*, *W*. *confusa*, and non-target strains. Clustering and PCA revealed a difference between some spectra of the three groups. Hierarchical clustering classified the spectra into two groups (Figure 2): one cluster included *W. confusa* and *W. cibaria,* and the other cluster included non-target species. In addition, PCA revealed clearly distinguishable clusters on the basis of the unique peptide fingerprint or main spectra library (MSP) of the two species (Figure 3). These results further show that MALDI-TOF MS can effectively differentiate among *W. cibaria, W. confusa*, and non-target species.

### 2.4. Characteristics of Mass Spectra

The species-specific mass-spectral loci that could classify the two species were explored. After the analysis, a matrix diagram was created on the basis of the presence or absence of mass peaks of different strains at a certain locus and their degree of difference (*p* < 0.001). Figure 4 presents the top 46 mass-spectral loci according to the difference between *W*. *cibaria* and *W*. *confusa*. Despite the spectra of the two *Weissella* species being similar, specific biomarkers were defined, allowing species-level discrimination. The spectral profiles of *W*. *cibaria* strains exhibited a specific peak at 6631.79 *m*/*z* (present in all strains except one) and a characteristic peak at 5352.49 *m*/*z*, which was present in 38 of the 44 strains analyzed. In contrast, the *W*. *confusa* species exhibited a specific and reproducible mass peak at 9682.98 *m*/*z*, which was present in all strains except for one. In addition, a characteristic peak at 3672.89 *m*/*z* was observed in 40 of the 46 strains analyzed. These results suggest that a combination of biomarkers can be used to distinguish between the two species. The peaks observed in bacteria through MALDI-TOF MS analysis are typically associated with ribosomal proteins, suggesting that these peaks may correspond to ribosomal proteins [21]. Nonetheless, as is consistent with prior studies, the precise identification of individual peaks remains unconfirmed [22,23,24].

### 2.5. Results of the Machine Learning Classification Model

MALDI-TOF MS analysis of 125 *Weissella* strains generated 500 protein fingerprints. These protein fingerprints (*n* = 500) contained 31,723 spectra, out of which 167 mass spectra were selected for quality control, smoothing, baseline correction, intensity calibration, spectra alignment, peak detection, and peak binning processes using the MALDIquant open-source package. Machine-learning models including ANN, PCA-KNN, and SVM were employed to discriminate among the two species and non-target samples. Results were obtained for training and test sets. Before the machine-learning models were developed, a correlation analysis was performed using the 167 *m*/*z* data. This analysis identified four bands with positive correlation of 0.60 or higher and approximately five bands with a negative correlation of −0.70 or higher. These bands can be used as indicators of classification feasibility (Figure 5).

For ANN, which was the first machine-learning algorithm, the training and test set accuracies were 0.997 and 0.978, respectively, which indicated that the three classes were generally well classified (Table 1). For PCA-KNN, PCA was used to reduce 167 dimensions, and the graph of the variance ratio of the principal components was examined. When six loadings were used, the variance ratio exceeded the 95% threshold, and data for these six components were used for developing the KNN (5) classification model. The results exhibited training- and test-set accuracies of 0.954 and 0.941, respectively. For Random Forest, the training- and test-set accuracies were 0.987 and 0.975, respectively.

The SVM used two soft kernels, namely, sigmoid and Radial Basis Function (RBF). When the sigmoid function was used, the training- and test-set accuracies were 0.973 and 0.904, respectively. In the RBF-based SVM model, the classification results were 1.0 for the training and test sets, thereby exhibiting the best results in differentiating the three groups (*W*. *cibaria*, *W*. *confusa*, and non-target). Moreover, the SVM–RBF algorithm consistently produced the same results when the three groups were classified in pairs. All three pairs, namely classifying *W*. *cibaria* and *W*. *confusa*, *W*. *cibaria* and non-target, and *W*. *confusa* and non-target, achieved a classification accuracy of 1.0. The developed models have been further supplemented with an analysis of Receiver Operating Characteristic (ROC) and Area Under the ROC Curve (AUC), as documented in Appendix A.

## 3. Discussion

MALDI-TOF MS is commonly used for bacterial identification in clinical microbiology laboratories. It relies on the detection of housekeeping and ribosomal proteins. This technique has been used for many decades and offers several advantages, such as being easy to use, fast, high throughput, cost effective, and reliable [25,26,27]. In our previous study, we used MALDI-TOF MS for identifying *Weissella* species and demonstrated that this technique, with an in-house database, can accurately identify *Weissella* species [10]. Integrating machine learning with MALDI-TOF MS enables highly accurate identification of closely-related species poorly discriminated in the database. Despite this potential, to our knowledge, no mass-peak-based machine-learning studies have been conducted for *Weissella* identification. Thus, this study investigated whether mass spectrometry techniques combined with machine-learning approaches could allow rapid species identification and accurate discrimination of closely related *Weissella* species.

In this study, to evaluate the potential of MALDI-TOF MS for differentiating among *W*. *cibaria*, *W*. *confusa*, and non-target strains, the correlation between the two species and mass spectra was analyzed. The commercial database did not accurately distinguish the two species; hence, developing machine-learning models for accurate identification and discrimination is necessary. Based on the mass spectra and concordance of the reference strain in the database, MALDI-TOF MS generates a report of the ten closest matches for each isolate [28], with matches arranged in the descending order of scores. MALDI-TOF MS results are usually interpreted according to the manufacturer’s recommendations, and these criteria are mostly sufficient for routine use in microbiology laboratories. However, interpreting the results is difficult, especially when different species belonging to the same genus having high scores are noted among the top ten best matches in the MALDI-TOF MS list [28,29]. Khot et al. (2012) proposed the 10% log-difference approach to address the problem of inconsistent results in the database [30]. This approach has been widely applied to strains having scores of ≥1.7, and consequently, 31 strains did not pass the cut-off. Further analysis based on single-colony identification identified these strains as different species (*W. cibaria* or *W. confusa*) with similar scores. In addition, our analysis revealed that ten strains were misidentified, which highlighted the challenge of distinguishing *W. cibaria* from *W. confusa* in the commercial database. These findings are consistent with those of previous studies that have also identified limitations with commercial databases [16,31,32]. Phylogenetically closely-related species often have similar protein fingerprinting, which makes it difficult to accurately discriminate between these species using commercial databases.

We analyzed 500 different spectra for *Weissella* strains, using PCA clustering and dendrogram techniques to classify them according to their protein fingerprinting. Our results, generated using MALDI-TOF MS fingerprinting, revealed that *W*. *cibaria* and *W*. *confusa* could be distinguished. Interestingly, the biomarker analysis revealed different loci of mass spectral peaks with *p* < 0.001. Among these loci, mass peaks at 6631.79, 5352.49, 9682.98, and 3672.89 *m*/*z* exhibited absolute specificity and may be potential biomarkers for distinguishing these two species. These findings highlight the ability of mass peaks to accurately identify *W*. *cibaria* and *W*. *confusa* among the *Weissella* species.

We employed machine-learning techniques to overcome the limitations of mass spectra analysis. Machine-learning algorithms can extract unique information from mass spectra, thus enabling more accurate analysis. Previous machine-learning studies conducted using MALDI-TOF MS data have detected antibiotic-resistant microorganisms [33]. To fully exploit the information in mass spectra obtained through MALDI-TOF MS, many researchers have implemented the machine-learning algorithm to simplify antimicrobial resistance determination [34,35]. Machine-learning methods can identify statistical dependencies in data and can consider nonlinear and interactive effects among features [20]. Therefore, these methods can reveal novel or unknown information present in MALDI-TOF MS that has been proven to be useful for antibiotic-resistance profiling in several studies [20]. Wang et al. (2022) used an SVM algorithm to differentiate carbapenem-resistant *Klebsiella pneumoniae* from carbapenem-sensitive strains based on the features obtained from MALDI-TOF MS [36]. SVM and the random-forest algorithm have been applied to the mass spectra of methicillin-resistant *Staphylococcus aureus* and beta-lactam-resistant *Bacteroides fragilis* for discriminating strains and for phenotypic prediction [37,38]. Compared with phenotypic-resistance profiling, their models have prediction accuracies of approximately 100% and significantly reduced the time required to initiate targeted antibiotic treatment. This study revealed that the information in MALDI-TOF MS can also aid in identifying and differentiating species, especially phylogenetically closely-related species. However, the disadvantage of this study is that only a small sample size was used to develop the machine learning model. Therefore, in future studies, it should be applied to larger samples associated with the characteristic peaks.

## 4. Materials and Methods

### 4.1. Isolation and Identification of W. cibaria and W. confusa

The type strains of *W*. *cibaria*, *W*. *confusa*, and non-target *Weissella* species were obtained from the Korean Collection for Type Culture (KCTC, Jeonju, Korea) and the Korean Agricultural Culture Collection (KACC, Jeonju, Korea) (Table 2). To isolate the strains from the samples, 19 food samples (kimchi, *n* = 11; jeotgal, *n* = 2, sikhae, *n* = 4, tempe = 2) were purchased from local markets. Each of these (25 g) was homogenized with 225 mL of 0.85% NaCl solution. The aliquots of the homogenized samples were serially diluted, and 0.1 mL of the diluted sample was spread on deMan, Rogosa, and Sharpe (MRS, MB cell, Seoul, Korea) agar plates, followed by incubation for 48 h at 30 °C. The colonies cultured on the plates were picked and streaked on fresh MRS agar (MB cell, Seoul, Korea), and aliquoted in 30% glycerol for storage at −80 °C. For the identification of the isolates, genomic DNA from *W*. *cibaria*, *W*. *confusa*, and non-target isolates was extracted using the DNeasy Blood and Tissue kit (Qiagen, Hilden, Germany) in accordance with the manufacturer’s instructions. Real-time PCR analysis was performed using a 7500 real-time PCR system (Applied Biosystems, Foster City, CA, USA) with specific primers listed in Table 3.

### 4.2. MALDI-TOF MS Analysis

#### 4.2.1. Protein Extraction

Proteins from *W*. *cibaria*, *W*. *confusa*, and non-target strains were extracted using the direct transfer-formic and ethanol-formic acid extraction protocol [39]. For the direct transfer-formic acid method, the fresh colony was smeared on a 96-position MALDI-TOF target plate (Bruker Daltonics, Bremen, Germany) in two replicates using a toothpick, overlaid with 1 µL of 70% formic acid, and then air dried at room temperature. Then, 1 µL of the HCCA matrix solution (α-cyano-4-hydroxycinnamic acid, Bruker Daltonics) dissolved in 50% acetonitrile, 2.5% trifluoroacetic acid, and 47.5% water was added to each of the spots and allowed to air dry. The on-plate protein extraction method was repeated twice to obtain spectra.

For the ethanol-formic acid-extraction procedure, a loopful of fresh bacterial culture (1 µL loopful of each isolate) was suspended in 300 µL of sterile pure water to which 900 µL of ethanol was added. The bacterial suspension was centrifuged at high speed (17,000× *g*) for 5 min; the supernatant was removed to completely discard the residual ethanol and recentrifuged. The resulting pellet was resuspended in 30 µL of 70% formic acid to which an equal volume of acetonitrile was added. After centrifugation at 17,000× *g* for 5 min, 1 µL of each supernatant was transferred to the 96-position MALDI-TOF target plate in two replicates, allowed to air dry, and then overlaid with 1 µL of the matrix solution. The spots were crystallized by air drying and then introduced into the instrument for MALDI-TOF MS analysis. The off-plate protein extraction method was repeated twice to obtain spectra.

#### 4.2.2. MALDI-TOF MS Analysis

The acquisition of mass spectra was performed using the Microflex LT mass spectrometer (Bruker Daltonics) operating in the automatic positive ion linear mode as the average of 240 laser shots. The spectra were collected over a mass range of 2000 and 20,000 Da using voltages of 20.00 kV (ion source 1), 18.70 kV (ion source 2), 9.15 kV (lens), and 1645 V (linear detector). The mass peak generated from each isolate was analyzed using BioTyper 3.0 software with reference database version 2022 (4274 species). The bacterial test standard (Bruker Daltonics) was used for calibration in accordance with the manufacturer’s instructions. The results were expressed using the criteria suggested by the manufacturer: scores of ≥2.0 indicate identification at the species level, scores of ≥1.7 but <2.0 indicate identification at the genus level, and scores <1.7 indicate unreliable identification.

### 4.3. Data Preprocessing for Machine Learning

#### 4.3.1. Data Set

All spectra data were classified into three groups for analysis, namely, “*W*. *cibaria* (*n* = 176)”, “*W*. *confusa* (*n* = 184)”, and “non-target (*n* = 140)” according to information about the strains. The data sets of “*W*. *cibaria*” and “*W*. *confusa*” were employed to explore the potential for the discrimination of *W*. *cibaria* and *W*. *confusa*; the datasets of “non-target” and “*W*. *cibaria*” were used to explore the potential for *W*. *cibaria* identification. The datasets of “non-target” and “*W*. *confusa*” were used to explore the potential for *W*. *confusa* identification.

#### 4.3.2. Data Preprocessing

The quality control and preprocessing of mass spectra were performed using the open-source MALDIquant module of the R software package version 4.3.1 created by statisticians Ross Ihaka and Robert Gentleman (Vienna, Austria). The mass spectra obtained using the MALDI-TOF MS instrument were exported as BrukerFlex files. These raw spectra were preprocessed using the R package “MALDIquant” module [40]. As a quality control, the raw spectra were tested to ensure that they contained the same number of data points and were not empty. The variance stabilization of raw spectra was performed using the square root transformation method. The mass spectra intensity was smoothed using “Savitzky–Golay–Filter” method with a halfWindowSize of 20. The SNIP algorithm was used for baseline correction and removal. Then, the intensity values were normalized using the Total-Ion-Current-Calibration method to allow a comparison of the peak intensities across the spectrum. Finally, the mass spectra were aligned by calibrating the mass values and aligning the spectra with the “alignSpectra” function. Before peak detection, the “averageMassSpectra” method was used to create a mean spectrum to average the technical replicates. Peak detection was then performed using the “MAD” method with a signal-to-noise ratio of 2. After the alignment process, the peak positions were quite similar but not identical. Therefore, binning was used to make the similar peak-position values identical (tolerance = 0.002). Finally, the peaks were filtered to retain as many features as possible and to remove false-positive peaks. The resulting data set was then scaled to a feature matrix for machine learning analysis.

#### 4.3.3. Proteomic Analysis

For proteomic analysis, the raw spectra preprocessed using MALDIquant were analyzed using Mass-Up: an open-source platform for analyzing MALDI data using machine learning and statistical techniques [41]. The mass spectra were transformed into mzML files (*m*/*z*-intensity lists), which enabled the implementation of a routine for the treatment of raw data. The mzML files were then imported into the statistical software Mass-Up version 1.0.14 created by López-Fernández et al. (Vigo, Spain) for the analysis of spectra data [41]. The quality control on the peak lists was performed using the MALDIquant method with a tolerance value of 0.002 ppm. Peak matching was performed using the MALDIquant method for intra- and inter-sample matching, considering a percentage of the presence of 50. Hierarchical clustering analysis and PCA were performed to evaluate the ability to discriminate between the two strains using mass spectra. A phyloproteomic tress was constructed using the “clustering analysis” function and default parameters. PCA was performed with the maximum number of components of −1 and a variance converted of 0.95. The biomarkers for the discrimination of the two species were identified using the “biomarker display” function as the mass peaks displaying the best *p* and q values.

### 4.4. Machine Learning Models for Classification

A machine-learning-based classification method was used to classify *W. cibaria*, *W. confusa*, and non-target strains. The classification method used machine-learning algorithms to analyze the spectral data obtained from MALDI-TOF MS. Specifically, it analyzed the intensity of 167 spectral peaks for each of the 500 samples. Four representative models were used for the machine-learning classification of this data. The x-input for each sample used in the training process included a sequence of 167 continuous data points. The samples were split randomly into 75% for training and 25% for testing.

The first machine-learning model used an artificial neural network with a structure consisting of two hidden layers. Each hidden layer comprised 200 and 125 neural network layers, respectively, and the ReLU function was used as the activation function. A softmax function was used to activate the output layer in this work, as per a common procedure when using a neural network to classify multiple classes. This method has led to these networks being labeled as ‘back propagation neural networks’ (BPNNs). While there exist unsupervised versions of artificial neural networks, the more frequently used approach involves the use of supervised settings for classification tasks with these networks. The number of classes was designated as L, and the total input of each unit $k=\left(1,\ldots ,\;K\right)$ in the output layer L was depicted as $u_{k}^{\left(L\right)}=W^{L}z^{\left(L-1\right)}+b^{\left(L\right)}$ based on the output of the immediate lower layer l=L−1. The output of the nth unit in the output layer was then obtained using Equation (1): the sum of the so-determined outputs (*y*) is always 1.
(1)$$y_{k}\equiv z_{k}^{\left(L\right)}=\frac{\exp\left(u_{k}^{\left(L\right)}\right)}{\sum_{j=1}^{K}\exp\left(u_{j}^{\left(L\right)}\right)}.$$

Training was performed using the Adam optimization function, and the model accuracy was evaluated after 50 epochs. The second machine-learning algorithm used PCA-KNN. The dimensionality of the 167 spectra was reduced through PCA, and only the influential features were selected to train the classification model. KNN is a popular supervised-learning technique owing to its simplicity and intuitiveness. By analyzing the neighboring data points, the KNN classifies a certain data point into a category with the highest number of similar data points. In this study, the value of K was determined based on the number of reduced dimensions obtained through PCA. The third classification-learning model used an SVM. SVMs are a type of supervised-learning algorithm that determines the optimal maximum margin hyperplane to distinguish between different classes within a higher-dimensional representation of the data instances. Essentially, a hyperplane refers to a plane that is one dimension less than the space it occupies. The SVM optimization process identifies the hyperplane that maximizes the distance or ‘gap’ between the plane and the instances. The conversion of data into this higher-dimensional space is facilitated by kernel functions, with the radial basis function kernel and the polynomial kernel being commonly utilized. Random Forest is a popular algorithm for machine learning employed in various fields, including the classification of spectral data. This methodology provides a robust mechanism for handling a vast number of predictors, many of which potentially could be irrelevant, rendering it particularly appropriate for spectral data where each spectral band can be considered a predictor. Due to these strengths, the Random Forest algorithm was chosen as the machine learning model for this study. Figure 6 depicts the schematic of these four models.

In addition to the accuracy analysis of the machine-learning models developed, the ROC curve analysis was conducted to compare the performance of the models. The ROC curve is a graphical plot that illustrates the diagnostic ability of a binary-classifier system, as its discrimination threshold is varied. The ROC curve is created by plotting the True Positive Rate (TPR) against the False Positive Rate (FPR) at various threshold settings. The TPR, also known as sensitivity, is a measurement of the proportion of actual positives that are correctly identified. Conversely, the FPR, or 1-specificity, measures the proportion of actual negatives that are incorrectly identified as positives. The AUC quantifies the overall ability of the model to distinguish between positive and negative classes. It provides a single scalar value that ranges from 0.5 to 1, where 0.5 signifies a model no better than random guessing, and 1 represents a perfect classifier. The larger the AUC, the better the model is at distinguishing between the positive and negative classes.

## 5. Conclusions

In conclusion, we, for the first time, demonstrated that MALDI-TOF MS combined with machine learning can discriminate among *W*. *cibaria*, *W*. *confusa*, and non-target strains. Furthermore, the model established using the SVM–RBF algorithm exhibited high reliability (accuracy of 1.0) in distinguishing among *W*. *cibaria*, *W*. *confusa*, and non-target strains. The fact that specific mass-to-charge (*m*/*z*) data obtained from protein fingerprints can act as potential mass speaks for reliably identifying *W*. *cibaria* and *W*. *confusa*. This approach can be implemented in food microbiology laboratories or clinical diagnostics industries to enable rapid, accurate, and cost-effective identification of *Weissella* species, thereby supplementing time consuming and costly conventional biochemical tests and sequencing methods. In addition, this approach can accurately distinguish *Weissella* species that are indistinguishable using the current MALDI-TOF MS database. It is envisioned that in the future, a machine-learning model integrated with mass spectra data will be employed for the classification of microbial species. This innovative combination of techniques holds great potential for advancing the field of microbial identification.

## Figures and Tables

**Figure 1 ijms-24-11009-f001:**
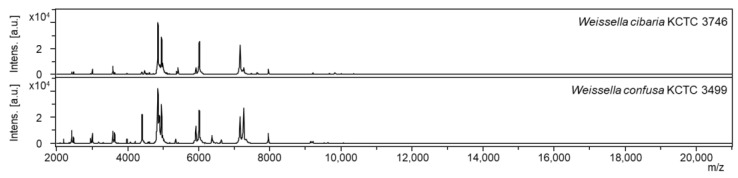
Mass spectra of *W*. *cibaria* KCTC 3746 and *W*. *confusa* KCTC 3499; *m*/*z*, mass-to-charge ratio; a.u., arbitrary units.

**Figure 2 ijms-24-11009-f002:**
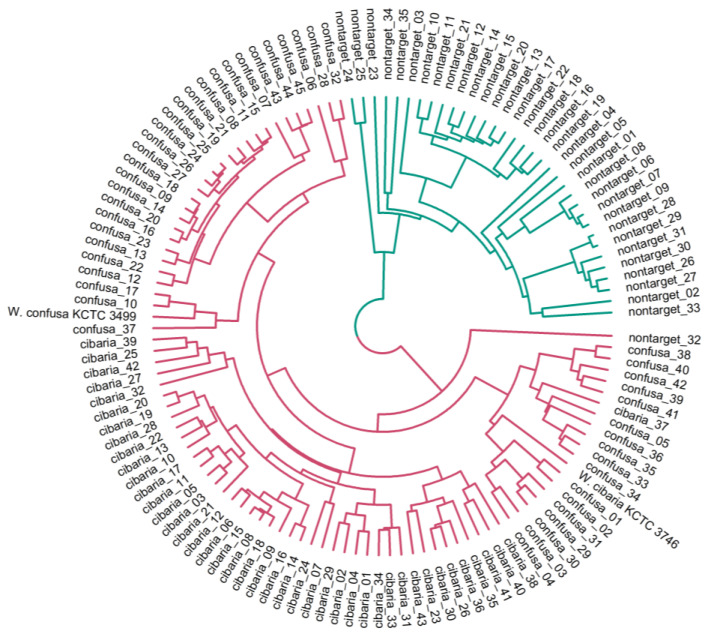
The hierarchical clustering based on the spectra of 44 *W*. *cibaria* strains, 46 *W*. *confusa* strains, and 35 non-target strains. The blue and red lines represent non-target and *W*. *cibaria* or *W*. *confusa* strains, respectively.

**Figure 3 ijms-24-11009-f003:**
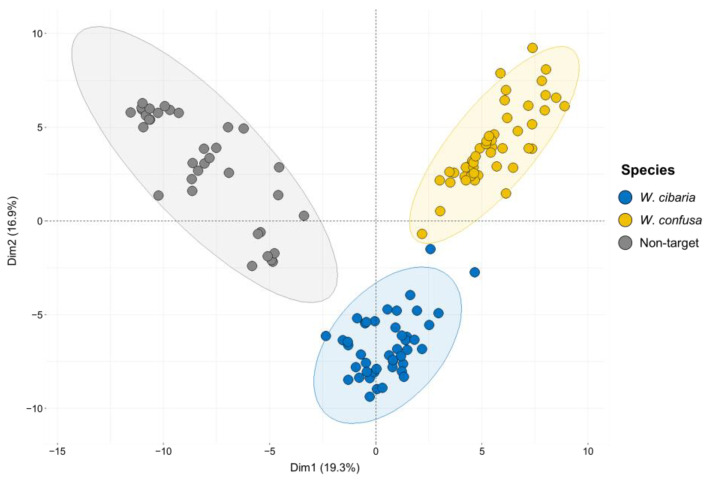
Principal component analysis (PCA) based on the spectra of 44 *W*. *cibaria* strains, 46 *W*. *confusa* strains, and 35 non-target strains. Each dot indicates average spectra per strain. Blue, yellow, and gray points represent *W*. *cibaria*, *W*. *confusa*, and non-target strains.

**Figure 4 ijms-24-11009-f004:**
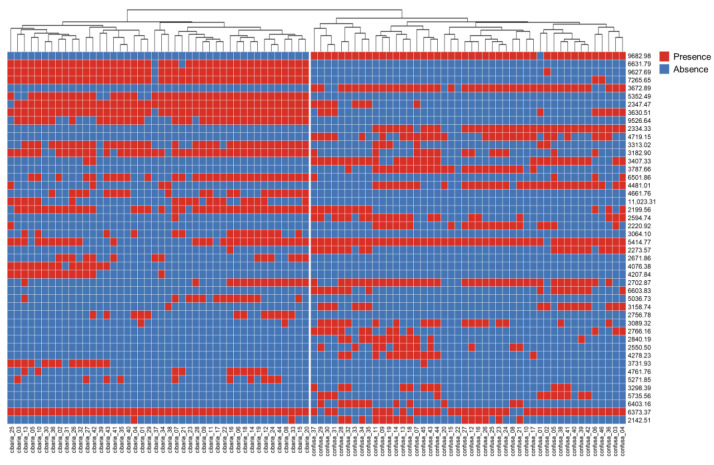
The matrix of absence/presence of mass-spectral loci among *W*. *cibaria* and *W*. *confusa* strains. The mass-spectral loci according to the difference (*p* < 0.001) between *W*. *cibaria* and *W*. *confusa* are listed on the right side of the matrix.

**Figure 5 ijms-24-11009-f005:**
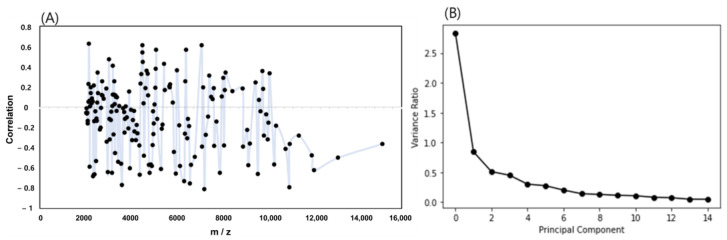
(**A**) Correlation between the *m*/*z* spectra peak and the classification of three groups, (**B**) scree plot to confirm the principal component threshold.

**Figure 6 ijms-24-11009-f006:**
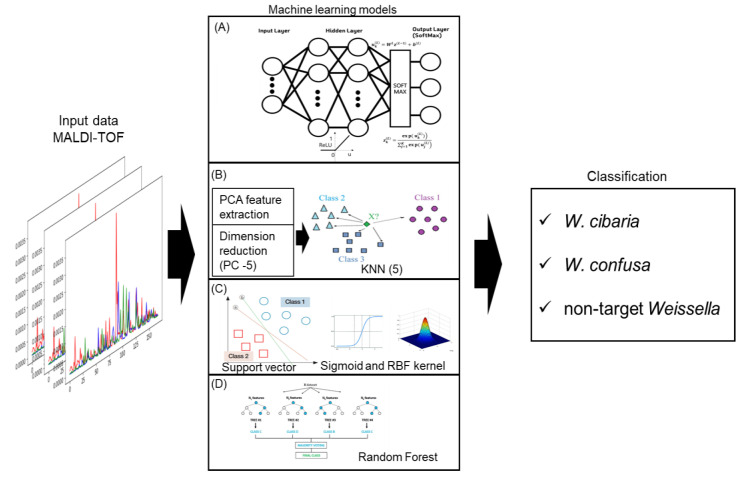
Overview of the four machine-learning models applied in this study; (**A**) artificial neural network structure with softmax function classifier applied; (**B**) PC-KNN with principal component analysis and nearest neighbor method applied; (**C**) SVM using sigmoid and RBF kernel; (**D**) Random Forest for classification model.

**Table 1 ijms-24-11009-t001:** Classification accuracy of the training set and test set of the machine learning algorithm used to classify the three groups.

Machine Learning Models	Training Set Accuracy	Test Set Accuracy
ANN	0.997	0.978
PCA-KNN	0.954	0.941
SVM-sigmoid	0.973	0.904
SVM-RBF	1	1
Random Forest	0.987	0.975

**Table 2 ijms-24-11009-t002:** Strain information used in this study.

Bacterial Strains	Origins
Reference strains	
*W. cibaria* KCTC ^1^ 12,413	-
*W. confusa* KCTC 3109	-
*W. ceti* KACC ^2^ 18,534	-
*W. cryptocerc* KACC 18,423	-
*W. distrammennae* KACC 16,890	-
*W. halotolerans* KACC 11,843	-
*W. hellenica* KACC 11,842	-
*W. koreensis* KACC 11,853	-
*W. minor* KCTC 3604	-
*W. paramesenteroides* KCTC 3531	-
*W. soli* KCTC 3789	-
*W. thailandensis* KCTC 3751	-
*W. uvarum* KACC 18,587	-
*W. viridescens* KCTC 3504	-
Isolates (*n* ^3^)	
*W*. *cibaria* (43)	Kimchi, sikhae, jeotgal
*W*. *confusa* (45)	Kimchi, jeotgal, tempe
*W*. *hellenica* (4)	Kimchi
*W*. *koreensis* (12)	Kimchi
*W*. *paramesenteroides* (7)	Tempe

^1^ KCTC, the Korean Collection for Type Cultures, ^2^ KACC, the Korean Agricultural Culture Collection, ^3^ Number of isolates.

**Table 3 ijms-24-11009-t003:** Specific primer information used in this study.

Target	Primer	Sequence (5′-3′)	Size (bp)	References
*W. cibaria*	CI-F	TGA ATC GTA CCT GAA GGA GC	99	[9]
	CI-R	TGA TAC TTT GCA AAC AGG CG		
*W. confusa*	CO-F	GTG CCG TAC TTC CAT GAC TT	177	[9]
	CO-R	GAC ATA CTT AAT GCC ATG TTC TGA C		

## Data Availability

The data presented in this study are available on request from the corresponding author.

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
