# Peer review of "Differentiation between Weissella cibaria and Weissella confusa Using Machine-Learning-Combined MALDI-TOF MS"

_ijms, 2023, doi:10.3390/ijms241311009_

Round 1
Reviewer 1 Report
The manuscript titled "Distinguishing between Weissella cibaria and Weissella confusa using a combined approach of machine learning and MALDI-TOF MS" presents a method that utilizes machine learning techniques in conjunction with MALDI-TOF MS to effectively differentiate between W. cibaria and W. confusa. This approach demonstrates considerable potential in overcoming the limitations of current taxonomic identification methods. However, before considering it for publication in the International Journal of Molecular Sciences, several issues need to be addressed.
1. The manuscript lacks mass spectra, making it challenging for readers to evaluate the dataset's quality, which is crucial for the analysis of mass spectrometry. Additionally, the repeatability of the quantitative data should be addressed.
2.. The identification of the specific ions used to distinguish the three species should be further explored and discussed, as this would enhance the manuscript.
3. The use of only 167 spectra for the machine learning models may not provide a sufficiently representative sample of the biological diversity and variability within these species. It is recommended to include larger datasets to obtain more generalizable results.
4. The study would benefit from a validation step using an independent dataset. While achieving perfect classification accuracy (1.0) is noteworthy, it could indicate overfitting, raising concerns about the model's ability to generalize to new data.
5. The manuscript mentions the application of multiple machine learning techniques (ANN, PCA-KNN, and SVM), but it lacks clarity on the rationale behind selecting these specific methods. It would be valuable to justify the chosen methods or conduct a comparative analysis of different machine learning techniques for this type of data.
6. The practical significance and real-world applications of the study should be further discussed. The authors should elaborate on how their findings can be implemented in microbial identification practices and consider the potential impact of their research on the field.
The manuscript's language could be improved for clarity and conciseness.
Author Response
Reviewer 1.
The manuscript titled "Distinguishing between Weissella cibaria and Weissella confusa using a combined approach of machine learning and MALDI-TOF MS" presents a method that utilizes machine learning techniques in conjunction with MALDI-TOF MS to effectively differentiate between W. cibaria and W. confusa. This approach demonstrates considerable potential in overcoming the limitations of current taxonomic identification methods. However, before considering it for publication in the International Journal of Molecular Sciences, several issues need to be addressed.
- The manuscript lacks mass spectra, making it challenging for readers to evaluate the dataset's quality, which is crucial for the analysis of mass spectrometry. Additionally, the repeatability of the quantitative data should be addressed.
Response: In this study, 500 protein fingerprints (31,723 spectra in total) were analyzed. A total of 167 mass spectra was obtained by aligning and quality checking 500 mass spectra, which were used to develop the machine learning models. The repeatability of the quantitative data was performed four times in total, two times with on-plate protein extraction and two times with off-plate protein extraction. As you recommended, we added the sentence in lines 273-274, 284-285, and 239-242 as follows:
Lines 273-274: The on-plate protein extraction method was repeated twice to obtain spectra.
Lines 284-285: The off-plate protein extraction method was repeated twice to obtain spectra.
Lines 239-242: The disadvantage of this study is that only a small sample size was used to develop the machine learning model. Therefore, in future studies, it should be applied to larger samples associated with the characteristic peaks.
2.. The identification of the specific ions used to distinguish the three species should be further explored and discussed, as this would enhance the manuscript.
Response: MALDI-TOF MS can be used to generate protein fingerprint signatures from whole bacterial cells. Using various algorithms, the bacteria can be rapidly identified by comparing these fingerprints with those in a database of reference spectra (Sogawa et al., 2011, Anal Bioanal Chem, 400). As you recommended, we added the sentence in lines 52-54 as follows:
Lines 52-54: MALDI-TOF MS can rapidly identify bacteria by generating protein fingerprint signatures from bacteria cells and comparing them to fingerprints in database of reference spectra [13].
- The use of only 167 spectra for the machine learning models may not provide a sufficiently representative sample of the biological diversity and variability within these species. It is recommended to include larger datasets to obtain more generalizable results.
Response: According to a previous study, the average spectra of strains to create machine learning models varied from 77 to 127 mass spectra (Delavy et al., 2020). We create machine learning models by selecting 167 mass spectra from a total of 31,723 spectra through a process according to the parameters of a previous study (Wang et al., 2022). Moreover, we isolated Weissella strains from various sources, such as kimchi, jeotgal, sikhae, and tempe. Previous studies have used 160 bacterial strains to develop a machine learning model for methicillin-resistant Staphylococcus aureus identification (Tang et al., 2021) and 115 bacterial strains for Riemerella anatipestifer model (Wang et al., 2022). We used a total of 125 Weissella strains. As you recommended, we added the sentence in lines 148-150 as follows:
Lines 148-150: MALDI-TOF MS analysis of 125 Weissella strains generated 500 protein fingerprints. These protein fingerprints (n = 500) contained 31,723 spectra, out of which 167 mass spectra were selected by data processing.
- The study would benefit from a validation step using an independent dataset. While achieving perfect classification accuracy (1.0) is noteworthy, it could indicate overfitting, raising concerns about the model's ability to generalize to new data.
Response: We appreciate your comments on the potential overfitting issue of the machine learning model in our paper. Initially, we classified the ionized waveforms from the protein mass analysis obtained from the MALDI-TOF MS equipment into three classes. As described in lines 153-155, there were about 9 peaks that had a correlation coefficient of over 0.6 in the classification correlation, and several other peaks also showed significance. This serves as an indicator that shows a good pattern for ensuring the accuracy we intend to distinguish through machine learning. In addition, in this study, we randomly split the data used for training and the data used for testing, ensuring that the test set consisted only of data not used for training. Thus, the ANN and PCA learning models had accuracies of 0.97 and 0.94 on the test set, while only the SVM-RBF had a high accuracy of 1.0 on the same test set, indicating that the models performed optimally and did not overfit on the dataset.
- The manuscript mentions the application of multiple machine learning techniques (ANN, PCA-KNN, and SVM), but it lacks clarity on the rationale behind selecting these specific methods. It would be valuable to justify the chosen methods or conduct a comparative analysis of different machine learning techniques for this type of data.
Response: Thank you for your comments. We have detailed the artificial neural network (ANN), PCA-KNN, and SVM machine learning methods used in this study, explaining the mechanisms and feature extraction methods of these three machine learning models to classify and recognize the peak data patterns of MALDI-TOF MS as well as other spectrometers (see lines 73-75, 355-358, and 372-379). We would also like to emphasize that machine learning models have been reported as potential tools for MALDI-TOF MS spectral analysis (Weis et al., 2020). As you recommended, we added the sentence in lines 73-75, 355-358, and 372-379 as follows:
Lines 73-75: Additionally, recent machine learning techniques have been utilized to maximize the use of information contained within MALDI-TOF MS, with the ultimate goal of improving species identification
Lines 355-358: This method has led to these networks being labeled as 'back propagation neural networks' (BPNNs). While there exist unsupervised versions of artificial neural networks, the more frequently used approach involves the use of supervised settings for classification tasks with these networks.
Lines 372-379: SVMs are a type of supervised learning algorithm that determines the optimal maximum margin hyperplane to distinguish between different classes within a higher-dimensional representation of the data instances. Essentially, a hyperplane refers to a plane that is one dimension less than the space it occupies. The SVM optimization process identifies the hyperplane that maximizes the distance or 'gap' between the plane and the instances. The conversion of data into this higher-dimensional space is facilitated by kernel functions, with the radial basis function kernel and the polynomial kernel being commonly utilized.
- The practical significance and real-world applications of the study should be further discussed. The authors should elaborate on how their findings can be implemented in microbial identification practices and consider the potential impact of their research on the field.
Response: As you recommended, we added the sentence in lines 390-394 as follows:
Lines 390-394: This approach can be implemented in food microbiology laboratories or clinical diagnostics industries for rapid, accurate, and cost-effective identification of Weissella species supplementing the conventional biochemical tests and sequencing. Also, a machine learning model combined with mass spectra data will be applied to the classification of microbial species in the future.
Reviewer 2 Report
This manuscript “Differentiation between Weissella cibaria and Weissella confus using machine learning-combined MALDI-TOF MS” by E. Kim et al, describes identification of two Weissella species by MALDI-TOF MS and machine learning techniques. Many of the related papers have been published in microbiology-related journals, and I recommend that this paper be submitted to a similar journal.
Author Response
Reviewer 2.
This manuscript “Differentiation between Weissella cibaria and Weissella confus using machine learning-combined MALDI-TOF MS” by E. Kim et al, describes identification of two Weissella species by MALDI-TOF MS and machine learning techniques. Many of the related papers have been published in microbiology-related journals, and I recommend that this paper be submitted to a similar journal.
Response: Thank you for your comments. This manuscript was submitted to the Special Issue "Advances in Mass Spectrometry-Based Proteomics" of the International Journal of Molecular Sciences. Prior to submission, we submitted an abstract to Richard Cao, Section Editor, and received permission for the topic to be accepted for publication.
Reviewer 3 Report
The article, "Differentiation between Weissella cibaria and Weissella confusa using machine learning-combined MALDI-TOF MS," submitted for review in the International Journal of Molecular Sciences, describes an experiment involving the use of artificial neural networks to differentiate and identify bacteria of two species: Weissella cibaria and Weissella confusa, based on the protein profile obtained by MALDI MS. The paper deals with an interesting topic and is part of the consideration of the insufficient discriminatory power of the MALDI MS method for identifying closely related bacterial species. Despite the well-designed study, the authors did not avoid several shortcomings that should be corrected before accepting the manuscript.
Major:
1. Section 2.4 presents m/z values that can serve as biomarkers to distinguish the bacterial species under study. What proteins correspond to these m/z values? Please do an ROC analysis for the proposed biomarkers (e.g., in the free online program metaboanalyst.ca).
2. The high classification accuracy values using machine learning described in Section 2.5 may be due to the selection of datasets described in Section 4.3.1. Using the same spectra as the dataset for creating the model and the test set may falsify the result. Please perform a similar analysis by dividing the spectra of W. cibaria and W. confusa between the teaching and test datasets, so that the spectra are not repeated in both datasets.
Author Response
Reviewer 3
The article, "Differentiation between Weissella cibaria and Weissella confusa using machine learning-combined MALDI-TOF MS," submitted for review in the International Journal of Molecular Sciences, describes an experiment involving the use of artificial neural networks to differentiate and identify bacteria of two species: Weissella cibaria and Weissella confusa, based on the protein profile obtained by MALDI MS. The paper deals with an interesting topic and is part of the consideration of the insufficient discriminatory power of the MALDI MS method for identifying closely related bacterial species. Despite the well-designed study, the authors did not avoid several shortcomings that should be corrected before accepting the manuscript.
Major:
- Section 2.4 presents m/z values that can serve as biomarkers to distinguish the bacterial species under study. What proteins correspond to these m/z values? Please do an ROC analysis for the proposed biomarkers (e.g., in the free online program metaboanalyst.ca).
Response: Thank you for your comments. The suggested free online program metaboanalyst.ca doesn’t work with our output data obtained by Bruker’s MALDI-TOF MS. Previous studies have used this program to find biomarkers by fusing MALDI-TOF MS with Fourier transform infrared (FTIR) spectroscopy or liquid chromatography tandem mass spectrometry (LC-MS/MS) data (Feng et al., 2020; Zhang et al., 2019). Since the peaks used for identification species of bacteria are ribosomal protein (Suarez et al., 2013), the species-specific peaks may belong to ribosomal protein. Many studies are unable to verify the identities of specific mass peaks (Pauker et al., 2018; Huang and Huang, 2018; Manzulli et al., 2021). As you recommended, we added sentence in lines 140-142 as follows:
Lines 140-142: The peaks in bacteria obtained from MALDI-TOF MS are assigned to ribosomal protein, so their peaks may belong to ribosomal protein [21]. However, similar to previous studies, the identity of each peak was not confirmed [22–24].
- The high classification accuracy values using machine learning described in Section 2.5 may be due to the selection of datasets described in Section 4.3.1. Using the same spectra as the dataset for creating the model and the test set may falsify the result. Please perform a similar analysis by dividing the spectra of W. cibaria and W. confusa between the teaching and test datasets, so that the spectra are not repeated in both datasets.
Response: Thank you for your comments regarding the concerns about the application of the developed learning model on new data. First, the ionized waveforms from the protein mass analysis obtained from the MALDI-TOF MS equipment were classified into three classes. This serves as an indicator that shows a good pattern for ensuring the accuracy we aim to distinguish through machine learning. In addition, in this study, we randomly split the data used for training and the data used for testing, ensuring that the test set consisted only of data not used for training. Thus, the ANN and PCA learning models had accuracies of 0.97 and 0.94 on the test set, while only the SVM-RBF had a high accuracy of 1.0 on the same test set, indicating that the models performed optimally and did not overfit on the dataset. We also note that Table 2 describes the diversity of the sources of the two strains used in this study.
Round 2
Reviewer 1 Report
The revisions and responses to the initial comments are appreciated. However, several areas might benefit from additional clarification or elaboration.
1. Even though the number of spectra and the repeatability of the experiment have been addressed, it remains unclear how the quality of these spectra was evaluated. Details regarding the quality control procedures implemented for the spectra prior to their usage in the machine learning models would be beneficial. Additionally, it would be highly beneficial if the authors could provide the MALDI-TOF MS spectra for the three species. This will offer additional insight into the proteomic differences between the species and strengthen the manuscript overall.
2. In relation to the MALDI-TOF MS technique, it's understood that it generates protein fingerprint signatures from whole bacterial cells. However, the initial comment was aimed more towards the identification of specific ions or mass-to-charge (m/z) ratios that are instrumental in distinguishing between W. cibaria and W. confusa. This information could be of great importance to both researchers and practitioners who aim to understand which specific proteins or peptides are essential for differentiating these species.
3. The response regarding the size of the dataset used for creating the machine learning models is in alignment with previous literature. Yet, the variability within the 167 spectra selected for the models remains somewhat ambiguous. Clarity on the selection criteria or process for these particular spectra from the original pool of 31,723 would be beneficial. Were these spectra chosen at random, or was there a specific selection criterion?
4. The discussion concerning the validation step of the machine learning models and the potential issue of overfitting is appreciated. Separating training and test sets is indeed critical for ensuring a fair evaluation of the model. Nevertheless, using an independent validation set, not involved in the model building process, is also a standard practice to assure the model's generalizability to unseen data. This practice might be worth considering in future studies.
5. It is commendable that the machine learning techniques utilized in this study are detailed, and sound reasons for their selection are provided. However, it would be insightful if an explanation were provided regarding why other common machine learning techniques (e.g., Random Forest, Gradient Boosting) were not included in this study. Were there specific reasons these models were not considered?
6. The added text about the real-world applications of this research is insightful. However, a stronger emphasis on the broader impact of this research could be beneficial, particularly regarding how this method can enhance current microbial identification techniques and the potential implications it may have for food safety and public health.
Minor editing of English language required.
Author Response
- Even though the number of spectra and the repeatability of the experiment have been addressed, it remains unclear how the quality of these spectra was evaluated. Details regarding the quality control procedures implemented for the spectra prior to their usage in the machine learning models would be beneficial. Additionally, it would be highly beneficial if the authors could provide the MALDI-TOF MS spectra for the three species. This will offer additional insight into the proteomic differences between the species and strengthen the manuscript overall.
Response: As you recommended, we revised the method to evaluate the quality of these spectra. Also, we added MALDI-TOF MS spectra for type strains of W. cibaria and W. confusa.
Lines 318-336: The quality control and preprocessing of mass spectra were performed using the MALDIquant open-source package available in the R software. The mass spectra obtained using the MALDI-TOF MS instrument were exported as BrukerFlex files. These raw spectra were preprocessed by the R package “MALDIquant” [40]. As a quality control, the raw spectra were tested to ensure that they contained the same number of data points and were not empty. The variance stabilization of raw spectra was performed using the square root transformation method. The mass spectra intensity was smoothed using “Savitzky–Golay–Filter” method with a halfWindowSize of 20. The SNIP algorithm was used for baseline correction and removal. Then, the intensity values were normalized using Total-Ion-Current-Calibration method to allow a comparison of the peak intensities across the spectrum. Finally, the mass spectra were aligned by calibrating the mass values and aligning the spectra with the "alignSpectra" function. Before peak detection, the “averageMassSpectra” method was used to create a mean spectrum to average the technical replicates. Peak detection was then performed using “MAD” method with a signal-to-noise ratio of 2. After alignment process, the peak positions were quite similar but not identical. Therefore, binning was used to make the similar peak position values identical (tolerance = 0.002). Finally, the peaks were filtered to retain as many features as possible and remove false positive peaks. The resulting data set was then scaled to a feature matrix for machine learning analysis.
Lines 109-111: The mass spectra of W. cibaria and W. confusa showed similar patterns (Figure 1). The mass spectra of each strain for non-target Weissella species are shown in Supplementary Figure S1.
Lines 428-429: Figure S1: Mass spectra of reference strains of non-target Weissella species; m/z, mass-to-charge ratio; a.u., arbitrary units.
Figure 1: We newly added the mass spectra of W. cibaria and W. confusa to Figure 1.
Figure S1: We newly added the mass spectra of non-target Weissella species to Figure S1.
- In relation to the MALDI-TOF MS technique, it's understood that it generates protein fingerprint signatures from whole bacterial cells. However, the initial comment was aimed more towards the identification of specific ions or mass-to-charge (m/z) ratios that are instrumental in distinguishing between W. cibaria and W. confusa. This information could be of great importance to both researchers and practitioners who aim to understand which specific proteins or peptides are essential for differentiating these species.
Response: Thank you for your comments. Since the peaks used for identification species of bacteria in MALDI-TOF MS are ribosomal protein (Suarez et al., 2013), the species-specific peaks may belong to ribosomal protein. Many studies are unable to verify the identities of specific mass peaks (Pauker et al., 2018; Huang and Huang, 2018; Manzulli et al., 2021). As you recommended, we revised the sentence in lines 20, 144-147, and 417-418 as follows:
Line 20: which led to the listing of potential species-specific mass-to-charge (m/z) loci.
Lines 144-147: The peaks observed in bacteria through MALDI-TOF MS analysis are typically associated with ribosomal proteins, suggesting that these peaks may correspond to ribosomal proteins [21]. Nonetheless, as is consistent with prior studies, the precise identification of individual peaks remains unconfirmed [22–24].
Lines 417-418: The specific mass-to-charge (m/z) obtained from protein fingerprint can act as potential mass speaks for reliably identifying W. cibaria and W. confusa.
- The response regarding the size of the dataset used for creating the machine learning models is in alignment with previous literature. Yet, the variability within the 167 spectra selected for the models remains somewhat ambiguous. Clarity on the selection criteria or process for these particular spectra from the original pool of 31,723 would be beneficial. Were these spectra chosen at random, or was there a specific selection criterion?
Response: The 167 spectra were selected from original pool for 31,723 spectra through a quality control, smoothing, baseline correction, intensity calibration, spectra alignment, peak detection, and peak binning processes using the MALDIquant open-source package. W. cibaria and W. confusa had very similar protein fingerprint, and most of the 31,723 spectra had the same mass-to-charge (m/z) values. As you recommended, we revised the sentence in lines 154-157 as follows:
Lines 154-157: out of which 167 mass spectra were selected for quality control, smoothing, baseline correction, intensity calibration, spectra alignment, peak detection, and peak binning processes using the MALDIquant open-source package.
- The discussion concerning the validation step of the machine learning models and the potential issue of overfitting is appreciated. Separating training and test sets is indeed critical for ensuring a fair evaluation of the model. Nevertheless, using an independent validation set, not involved in the model building process, is also a standard practice to assure the model's generalizability to unseen data. This practice might be worth considering in future studies.
Response: Thank you for your comments. Establishing standard practices to ensure the generalizability of models to unseen data is important and worthy of further consideration in future research.
- It is commendable that the machine learning techniques utilized in this study are detailed, and sound reasons for their selection are provided. However, it would be insightful if an explanation were provided regarding why other common machine learning techniques (e.g., Random Forest, Gradient Boosting) were not included in this study. Were there specific reasons these models were not considered?
Response: Thank you for your comments. As you recommended, we have updated our manuscript to include the performance of the Random Forest classifier, with relevant changes made to lines 389-394, Figure 6, and Table 1. In our future research, we intend to utilize Gradient Boosting algorithms. In addition, we described in the introduction the machine learning models we implemented, which are commonly used in MALDI-TOF spectrum analysis.
Line 22: and random forest were used.
Lines 173-174: For Random Forest, the training and test set accuracies were 0.987 and 0.975, respectively.
Lines 389-394: Random Forest is a popular algorithm for machine learning employed in various fields, including the classification of spectral data. This methodology provides a robust mechanism for handling a vast number of predictors, many of which could potentially be irrelevant, rendering it particularly appropriate for spectral data where each spectral band can be considered a predictor. Due to these strengths, the Random Forest algorithm was chosen as the machine learning model for this study.
Lines 397-400: Overview of the four machine learning models applied in this study; (A) artificial neural network structure with softmax function classifier applied; (B) PC-KNN with principal component analysis and nearest neighbor method applied; (C) SVM using sigmoid and RBF kernel; (D) Random Forest for classification model
Figure 6: We newly added random forest analysis to Figure 6.
Table 1: We newly added results for random forest analysis to Table 1.
- The added text about the real-world applications of this research is insightful. However, a stronger emphasis on the broader impact of this research could be beneficial, particularly regarding how this method can enhance current microbial identification techniques and the potential implications it may have for food safety and public health.
Response: As you recommended, we added the sentence in lines 418-426 as follows:
Lines 418-426: This approach can be implemented in food microbiology laboratories or clinical diagnostics industries to enable rapid, accurate, and cost-effective identification of Weissella species, thereby supplementing time consuming and costly conventional biochemical tests and sequencing methods. In addition, this approach can accurately distinguish Weissella species that are indistinguishable with the current MALDI-TOF MS database. It is envisioned that in the future, a machine learning model integrated with mass spectra data will be employed for the classification of microbial species. This innovative combination of techniques holds great potential for advancing the field of microbial identification.
Reviewer 2 Report
Since MS is essentially a detection method with low reproducibility, reproducibility data from this study should be added.
OK.
Author Response
Since MS is essentially a detection method with low reproducibility, reproducibility data from this study should be added.
Response: Thank you for your comments. In this study, 500 protein fingerprints (31,723 spectra in total) were analyzed. A total of 167 mass spectra was obtained by aligning and quality checking 500 mass spectra, which were used to develop the machine learning models. The repeatability of the quantitative data was performed four times in total, two times with on-plate protein extraction and two times with off-plate protein extraction. The reproducibility data from this study is mentioned on lines 153-157, 283-284, and 294-295. As you recommended, we added MALDI-TOF MS spectra for type strains of W. cibaria and W. confusa.
Lines 153-157: MALDI-TOF MS analysis of 125 Weissella strains generated 500 protein fingerprints. These protein fingerprints (n = 500) contained 31,723 spectra, out of which 167 mass spectra were selected for quality control, smoothing, baseline correction, intensity calibration, spectra alignment, peak detection, and peak binning processes using the MALDIquant open-source package.
Lines 283-284: The on-plate protein extraction method was repeated twice to obtain spectra.
Lines 294-295: The off-plate protein extraction method was repeated twice to obtain spectra.
Lines 109-111: The mass spectra of W. cibaria and W. confusa showed similar patterns (Figure 1). The mass spectra of each strain for non-target Weissella species are shown in Supplementary Figure S1.
Figure 1: We newly added the mass spectra of W. cibaria and W. confusa to Figure 1.
Figure S1: We newly added the mass spectra of non-target Weissella species to Figure S1.
Reviewer 3 Report
I thank the authors for their responses.
- Section 2.4 presents m/z values that can serve as biomarkers to distinguish the bacterial species under study. What proteins correspond to these m/z values? Please do an ROC analysis for the proposed biomarkers (e.g., in the free online program metaboanalyst.ca).
Response: Thank you for your comments. The suggested free online program metaboanalyst.ca doesn’t work with our output data obtained by Bruker’s MALDI-TOF MS. Previous studies have used this program to find biomarkers by fusing MALDI-TOF MS with Fourier transform infrared (FTIR) spectroscopy or liquid chromatography tandem mass spectrometry (LC-MS/MS) data (Feng et al., 2020; Zhang et al., 2019).
I disagree with the authors. The metaboanalyst.ca program has the ability to analyze data from Bruker instruments as well. Data can be saved and analyzed in .txt format. This program has also been used many times to analyze biomarkers also from data derived from LDI spectra, for example: doi:10.1016/j.jpba.2020.113752; 10.1007/s00216-020-02807-1. There are certainly other programs that allow you to generate ROC plots. I still maintain that this analysis should be added to the manuscript.
Since the peaks used for identification species of bacteria are ribosomal protein (Suarez et al., 2013), the species-specific peaks may belong to ribosomal protein. Many studies are unable to verify the identities of specific mass peaks (Pauker et al., 2018; Huang and Huang, 2018; Manzulli et al., 2021). As you recommended, we added sentence in lines 140-142 as follows:
Lines 140-142: The peaks in bacteria obtained from MALDI-TOF MS are assigned to ribosomal protein, so their peaks may belong to ribosomal protein [21]. However, similar to previous studies, the identity of each peak was not confirmed [22–24].
Thank you for the added explanation.
- The high classification accuracy values using machine learning described in Section 2.5 may be due to the selection of datasets described in Section 4.3.1. Using the same spectra as the dataset for creating the model and the test set may falsify the result. Please perform a similar analysis by dividing the spectra of W. cibaria and W. confusa between the teaching and test datasets, so that the spectra are not repeated in both datasets.
Response: Thank you for your comments regarding the concerns about the application of the developed learning model on new data. First, the ionized waveforms from the protein mass analysis obtained from the MALDI-TOF MS equipment were classified into three classes. This serves as an indicator that shows a good pattern for ensuring the accuracy we aim to distinguish through machine learning. In addition, in this study, we randomly split the data used for training and the data used for testing, ensuring that the test set consisted only of data not used for training. Thus, the ANN and PCA learning models had accuracies of 0.97 and 0.94 on the test set, while only the SVM-RBF had a high accuracy of 1.0 on the same test set, indicating that the models performed optimally and did not overfit on the dataset. We also note that Table 2 describes the diversity of the sources of the two strains used in this study.
Thank you and accept the answer.
Author Response
- I disagree with the authors. The metaboanalyst.ca program has the ability to analyze data from Bruker instruments as well. Data can be saved and analyzed in .txt format. This program has also been used many times to analyze biomarkers also from data derived from LDI spectra, for example: doi:10.1016/j.jpba.2020.113752; 10.1007/s00216-020-02807-1. There are certainly other programs that allow you to generate ROC plots. I still maintain that this analysis should be added to the manuscript.
Response: Thank you for your comments. As you recommended, we have conducted ROC curve analysis for the developed classification models. We propose to include these results as supplementary data for further consideration.
Lines 182-184: The developed models have been further supplemented with an analysis of Receiver Operating Characteristic (ROC) and Area Under the ROC Curve (AUC), as documented in Supplementary Figure S2‒S6.
Lines 401-411: In addition to the accuracy analysis of the machine learning models developed, the ROC curve analysis was conducted to compare the performance of the models. The ROC curve is a graphical plot that illustrates the diagnostic ability of a binary classifier system as its discrimination threshold is varied. The ROC curve is created by plotting the True Positive Rate (TPR) against the False Positive Rate (FPR) at various threshold settings. The TPR, also known as sensitivity, is a measurement of the proportion of actual positives that are correctly identified. Conversely, the FPR, or 1-specificity, measures the proportion of actual negatives that are incorrectly identified as positives. The AUC quantifies the overall ability of the model to distinguish between positive and negative classes. It provides a single scalar value that ranges from 0.5 to 1, where 0.5 signifies a model no better than random guessing, and 1 represents a perfect classifier. The larger the AUC, the better the model is at distinguishing between the positive and negative classes.
Lines 429-447: Figure S2: The developed ANN model was utilized to analyze the ROC curves and AUC, showing the following: (Left) The classification performance between W. cibaria and the other two classes, (middle) The classification performance between W. confusa and the other two classes, (Right) The classification performance between non-target samples and the other two classes. Figure S3: The developed PCA-KNN model was utilized to analyze the ROC curves and AU, showing the following: (Left) The classification performance between W. cibaria and the other two classes, (middle) The classification performance between W. confusa and the other two classes, (Right) The classification performance between non-target samples and the other two classes. Figure S4: The developed SVM-sigmoid model was utilized to analyze the ROC curves and AUC, showing the following: (Left) The classification performance between W. cibaria and the other two classes, (middle) The classification performance between W. confusa and the other two classes, (Right) The classification performance between non-target samples and the other two classes. Figure S5: The developed Random Forest model was utilized to analyze the ROC curves and AUC, showing the following: (Left) The classification performance between W. cibaria and the other two classes, (middle) The classification performance between W. confusa and the other two classes, (Right) The classification performance between non-target samples and the other two classes. Figure S6: The developed Sigmoid-RBF model was employed for the analysis of ROC curves and AUC, achieving a classification accuracy of 1.0 across all three categories.
Figure S2-S6: We newly added ROC curves to Figure S2-S6.
Round 3
Reviewer 1 Report
I am overall satisfied with the addressed points and the changes made. The authors have taken into account my comments, providing detailed and relevant answers, as well as making appropriate revisions to the manuscript. The manuscript has seen significant improvement, and I believe it is suitable for publication now.
Minor editing of English language required.
Author Response
I am overall satisfied with the addressed points and the changes made. The authors have taken into account my comments, providing detailed and relevant answers, as well as making appropriate revisions to the manuscript. The manuscript has seen significant improvement, and I believe it is suitable for publication now.
Response: Thank you for your comments.
Reviewer 2 Report
Response: Thank you for your comments. In this study, 500 protein fingerprints (31,723 spectra in total) were analyzed. A total of 167 mass spectra was obtained by aligning and quality checking 500 mass spectra, which were used to develop the machine learning models. The repeatability of the quantitative data was performed four times in total, two times with on-plate protein extraction and two times with off-plate protein extraction.
Thanks for your response.
I am not sure why and how you choose 167 from 500 mass spectra. All should be included for analysis.
Author Response
Reviewer 2.
I am not sure why and how you choose 167 from 500 mass spectra. All should be included for analysis.
Response: The 167 spectra were selected from original pool for 31,723 spectra through a quality control, smoothing, baseline correction, intensity calibration, spectra alignment, peak detection, and peak binning processes using the MALDIquant open-source package. W. cibaria and W. confusa had very similar protein fingerprint, and most of the 31,723 spectra had the same mass-to-charge (m/z) values. The process of selecting 167 spectra is presented in lines 154-157 and 318-336 as follows:
Lines 154-157: out of which 167 mass spectra were selected for quality control, smoothing, baseline correction, intensity calibration, spectra alignment, peak detection, and peak binning processes using the MALDIquant open-source package.
Lines 318-336: The quality control and preprocessing of mass spectra were performed using the MALDIquant open-source package available in the R software. The mass spectra obtained using the MALDI-TOF MS instrument were exported as BrukerFlex files. These raw spectra were preprocessed by the R package “MALDIquant” [40]. As a quality control, the raw spectra were tested to ensure that they contained the same number of data points and were not empty. The variance stabilization of raw spectra was performed using the square root transformation method. The mass spectra intensity was smoothed using “Savitzky–Golay–Filter” method with a halfWindowSize of 20. The SNIP algorithm was used for baseline correction and removal. Then, the intensity values were normalized using Total-Ion-Current-Calibration method to allow a comparison of the peak intensities across the spectrum. Finally, the mass spectra were aligned by calibrating the mass values and aligning the spectra with the "alignSpectra" function. Before peak detection, the “averageMassSpectra” method was used to create a mean spectrum to average the technical replicates. Peak detection was then performed using “MAD” method with a signal-to-noise ratio of 2. After alignment process, the peak positions were quite similar but not identical. Therefore, binning was used to make the similar peak position values identical (tolerance = 0.002). Finally, the peaks were filtered to retain as many features as possible and remove false positive peaks. The resulting data set was then scaled to a feature matrix for machine learning analysis.